# Research on Interlayer Bonding Quality Control Method of Dam Concrete Based on Equivalent Age

**DOI:** 10.3390/ma14185192

**Published:** 2021-09-09

**Authors:** Wei-Jia Liu, You-Bo Wang, Qing-Bin Li, Xiao-Feng Gao, Yao-Shen Tan, Chun-Feng Liu, Yu Hu, Xu-Jing Niu

**Affiliations:** 1State Key Laboratory of Hydroscience and Engineering, Tsinghua University, Beijing 100084, China; liuweijia@tsinghua.edu.cn (W.-J.L.); wangyob@163.com (Y.-B.W.); qingbinli@tsinghua.edu.cn (Q.-B.L.); 2College of Civil Engineering, Zhejiang University of Technology, Hangzhou 310023, China; gaoxf@zjut.edu.cn; 3China Three Gorges Construction Engineering Corporation, Beijing 100000, China; tan_yaosheng@ctg.com.cn (Y.-S.T.); liu_chunfeng@ctg.com.cn (C.-F.L.); 4School of Mechanics and Civil Engineering, China University of Mining and Technology (Beijing), Beijing 100083, China

**Keywords:** dam concrete, interlayer properties, interlayer bonding quality, equivalent age, early-warning system

## Abstract

Interlayer bonding quality is the key to the stability and durability of dam concrete. In this study, interlayer splitting tensile strength, relative permeability coefficient, and electric flux of dam concrete at different temperatures were tested. The relationships between equivalent age and strength coefficient, relative permeability coefficient ratio, and electric flux ratio were established. Meanwhile, a comprehensive early-warning and control system of dam interlayer bonding quality based on the above relationships was proposed. The results showed that the interlayer mechanical properties, impermeability, and anti-chloride ion permeability of dam concrete decreased with the increase of temperature. Moreover, the equivalent age was linearly correlated with strength coefficient, relative permeability coefficient ratio, and electric flux ratio of concrete. The correlation coefficients were 0.986, 0.973, and 0.924, respectively. In addition, the interlayer bonding quality of dam concrete can be effectively controlled by the early-warning system established according to the relationship between equivalent age and interlayer properties parameters.

## 1. Introduction

The development of dam construction technology can be divided into artificial, mechanized, automatic, digital, and intelligent stage [1]. The intelligent dam proposes a closed-loop intelligent construction concept based on perception, analysis, and control. Intelligent dam construction technology includes intelligent concrete leveling [2], concrete vibration [3], spray [4], early warning (interlayer bonding properties and dam temperature) [5], water cooling [6], maintenance [7], grouting [8], etc. China’s dam construction has stepped into an intelligent era from the digital stage [9]. Intelligent construction theories and systems have been successfully applied to the construction of ultra-high arch dams, such as Xiluodu (285.5 m), Wudongde (277 m), and Baihetan (279 m).

The Wudongde and Baihetan dam construction areas are of a typical dry hot valley climate. Extreme weather, such as high temperature, low humidity, strong wind, and short-time heavy precipitation, occurs frequently in dam construction areas [10]. Under the coupling effect of strong wind and dry hot climate, the following problems will occur in the pouring process of concrete mixture: (1) The low bleeding and high evaporation of concrete affect the hydration process and final hydration degree of cement [11]; (2) Aggregates whitening and false setting of surface concrete due to water loss (see Figure 1a); (3) Loss of water will cause a sharp change in capillary negative pressure value of concrete, and then cause plastic cracking of concrete [12] (see Figure 1b); and (4) At high temperature, the early bleeding rate of concrete is greater than the evaporation rate [13], which will lead to a layer of floating slurry on the surface of concrete (see Figure 1c). The high water-cement ratio of floating slurry will make the concrete form sparse and porous interlayer weak zone [14]. It can be seen that the bad weather will lead to the deterioration of the state of concrete layer. If the above problems cannot be effectively solved, seepage channels may occur between layers. In addition, there may be cracks and anti-sliding stability problems.

Many studies have proved the existence of the above problems from macro and micro levels. Niu et al. [15] investigated the effects of ambient temperature and wind speed on splitting tensile strength, chloride ion penetration resistance, and pore structure of layered concrete. The results show that the increase of temperature or wind speed decreased interlayer splitting tensile strength and chloride ion penetration resistance, and the pore structure of the concrete becomes rough. Ribeiro et al. [16] studied the influence of relative humidity of the concrete surface on interlayer tensile strength. The results show that the effect of relative humidity on strength decreased with the increase of maturity. In addition to environmental factors, the interval time of concrete pouring also greatly affects interlayer performance. For example, Karimpour [17] found that the compressive strength and impermeability of RCC deteriorated with the increase of interval time. Researchers such as Qian [18] and Qin [19] have found that splitting tensile strength of mortar decreases and permeability coefficient increases with the increase of interlayer interval time. If interval time exceeds initial setting time, the pore structure of the concrete at the interface will tend to be connected and micro-cracks will appear, which will lead to the interlayer bond strength being lower than the strength of the bulk concrete [20]. Therefore, they suggest pouring the upper concrete before initial setting of the lower concrete.

The construction quality control of high arch dams covers many aspects, including the performance of raw materials, control of concrete temperature, operation of large machinery, pouring, and maintenance of concrete, etc. This research mainly focuses on the quality control of interlayer bonding in the concrete pouring process. The setting time method [21], maturity method, and water-strength relationship method [22,23] are commonly used in the construction of dams to control the interlayer bonding quality.

The setting time method determines the pouring time node of the upper layer concrete by controlling the setting state of the lower layer concrete. If interval time exceeds the initial setting time of the lower layer concrete, pouring should be stopped.

Maturity is a concept that has existed since 1950. At present, this method has been widely used in various fields, such as hydraulic engineering, construction, and transportation. In the field of construction engineering, the relationship between maturity and compressive strength was used to judge the time of the mold removal to avoid building collapse caused by premature demolding [24]. In the process of roads [25,26] and bridges [27] construction, the relationship between maturity and compressive strength was mainly used to judge the time node of the road opening to prevent damage to the road surface caused by premature opening. In addition, the maturity theory has also been preliminarily applied in the process of hydraulic engineering construction. Li et al. [28] studied the influence of ambient temperature on the fracture performance of cast-in-place dam concrete. The relationship between fracture parameters and equivalent maturity of concrete was established based on maturity theory. Ustabas [29] studied the feasibility of the weighted maturity method in predicting the compressive strength of dam concrete. The results show that the weighted maturity method can be used to estimate the compressive strength of dam concrete at lower hydration temperatures. In addition, the American Concrete Association Committee recommended that joint maturity be used to control the quality of RCC horizontal joints [30].

Xu et al. [22] tested the water content and interlayer mechanical properties of layered concrete under the influence of different environmental factors. Then, the prediction model of interlayer bonding strength was deduced based on the strength development model and real hydration degree of concrete. Finally, an interlayer bonding quality control method based on the moisture content of the lower layer was proposed. The interval time can be controlled by the concrete water content at the construction site.

The environmental factors of the construction site changed with time, which affected the hydration process of the cement. Therefore, it is difficult to accurately test the setting time of concrete. In addition, the relationship between setting state and the strength of the concrete has not been established. In conclusion, the application of setting time method in the field needs to be further studied. Although the maturity method establishes the relationship between concrete maturity and compressive strength, compressive strength is not the most direct index to reflect the interlayer bonding quality of concrete. Although the water-strength method established the relationship between the water index and interlayer mechanical properties, the applicability of this method in the case of concrete bleeding was not considered. In addition, as a water retaining structure, the interlayer bonding quality should be controlled comprehensively from the aspects of interlayer mechanical properties and impermeability. Because the interlayer mechanical properties and impermeability are the key indexes affecting the stability and durability of the dam.

The current dam construction process is very random and lacks a method that can automatically collect on-site construction information (such as meteorological data, concrete parameters, construction methods, etc.) and achieve interlayer bonding quality control. The interlayer bonding quality control method of dam concrete should establish the relationship between the key parameters of the lower concrete and interlayer performance to determine the specific construction node. In view of the shortcomings of above methods, the tests of interlayer splitting tensile strength, relative permeability coefficient, and electric flux of concrete at different temperatures were carried out, respectively. Then, the relationship between equivalent age and interlayer strength coefficient, relative permeability coefficient ratio, and electric flux ratio were established according to maturity theory. Finally, the control index of interlayer bonding quality was put forward according to the guaranteed rate of strength, permeability resistance, and chloride ion permeability resistance. In addition, a comprehensive control system of “monitoring-analysis-feedback-processing” has been established. This system realizes effective monitoring of quality parameters of the dam during concrete pouring and real-time warning of performance indexes. Furthermore, effective treatment measures were put forward for possible construction risks, which can guide the construction in concrete pouring and ensure the quality of interlayer bonding to meet the design requirements. The interlayer bonding quality control method proposed in this paper has been successfully applied in the Wudongde dam (see Figure 2a) and the Baihetan dam (see Figure 2b). The locations of the two dams are shown in Figure 3.

## 2. Experimental Procedure

### 2.1. Raw Materials

The physical properties and chemical components of cement were provided by the manufacturer, which are shown in Table 1. The mix proportion of concrete is shown in Table 2. The water-cement ratio is 1.07. The apparent density of artificial sand (crushed basalt) is 2780 kg/m^3^. The coarse aggregate is crushed basalt with a particle size range of 5–20 mm. In addition, polycarboxylic acid superplasticizer was used to maintain the fluidity of concrete, and the water reduction rate was about 19%.

### 2.2. Specimen Preparation

Concrete specimens were poured in three steps. First, the lower concrete (half height of the mold) was poured. Then, the specimens were placed in environmental chambers (temperatures of 20, 30, and 40 °C, and relative humidity of 30%). After 6 h, the specimens were taken out of the environmental chambers, and the upper layer of concrete was poured (Figure 4). All specimens were placed in an indoor environment (20 ± 2 °C, relative humidity 50%) for 24 h before demolding. After demolding, the specimens were put into the standard curing room (20 ± 2 °C, relative humidity of 95%). The properties of concrete were tested after 28 days.

There were three types of molds: cube (150 mm × 150 mm × 150 mm), truncated cone (upper diameter of 175 mm, lower diameter of 185 mm, height of 150 mm), and cylinder (ϕ100 mm × 200 mm). Cube specimens were used for splitting tensile strength test, truncated cone specimens were used for relative permeability coefficient test, and cylinder specimens were used for electric flux test. Six samples were prepared for each analysis.

The specimens were divided into four groups. The numbers of concrete specimens were C0, T20, T30, and T40, respectively. C0 represents the bulk concrete, and the other groups are layered concrete.

### 2.3. Experimental Methods

#### 2.3.1. Splitting Tensile Strength Test of Concrete

The splitting tensile strength test of concrete was carried out according to the Chinese standard DL/T 5150-2017 [31]. First, the specimen was placed in the center of the pressure plate under the pressure testing machine. Then cushion strips were placed between the upper and lower pressing plates and the test specimen (the placing direction of the cushion strip was horizontal with the layer, see Figure 5). The loading rate was 0.04~0.06 MPa/s. The calculation formula of splitting tensile strength of concrete is as follows:(1)$$f_{ts}=\frac{2P}{\pi A}=0.637\frac{P}{A}$$
where $f_{ts}$ is splitting tensile strength of concrete (MPa); P is peak load; and A is cross-sectional area of the cube specimen. The average of six test specimens was taken in each group.

#### 2.3.2. Concrete Relative Permeability Test

The relative permeability test of concrete was carried out according to the Chinese standard DL/T 5150-2017 [31]. The main purpose of the test was to determine the water seepage height of the concrete under constant water pressure and calculate the relative permeability coefficient. First, the specimens after standard curing for 28 d were placed in impermeable testing machines. Then, a stable pressure value of 0.8 MPa was loaded and maintained for 24 h. Finally, the specimen was split along the longitudinal section and the seepage height was recorded. The relative permeability coefficient is calculated by Equation (2).
(2)$$K_{r}=\frac{aD_{m}^{2}}{2tH}$$
where $K_{r}$ is the relative permeability coefficient, mm/h; a is the water absorption rate of concrete, generally 0.03; $D_{m}$ is the average height of water seepage, mm; t is constant pressure time, h; and H is water pressure, expressed by the height of water column, mm (1 MPa water pressure, expressed as 102,000 mm in water column height).

#### 2.3.3. Resistance to Chloride Ion Penetration Test (Electric Flux Method)

The rapid chloride ion penetration test of concrete was carried out in accordance with Chinese national standard GB/T 50082-2009 [32]. The direct current flux method is the standard test method for the resistance of concrete to chloride ion penetration. First, a 60 V direct current was applied between the electrodes, and the current was used at the beginning of the test for 1 min as initial current $I_{0}$. Then the test data were recorded every 30 min for 6 h. The calculation formula of electric flux is as follows:(3)$$Q=900(I_{0}+2I_{30}+2I_{60}+\ldots +2I_{t}+\ldots +2I_{300}+2I_{330}+I_{360})$$
where Q is total electric flux (C) passing through the specimen; $I_{0}$ is initial current (A); and $I_{t}$ is current at time t.

The electric flux test method is shown in Figure 6. Chloride ions move to the positive electrode through the concrete specimen under the action of DC voltage. Measuring the amount of electric charge passing through the concrete which can indirectly reflect the ability of the concrete to resist the penetration of chloride ions.

## 3. Results and Discussion

### 3.1. Interlayer Properties

#### 3.1.1. Splitting Tensile Strength

Figure 7 shows the change of interlayer splitting tensile strength of concrete at different temperatures. First, the interlayer mechanical properties decreased with the increase of temperature. The splitting tensile strength of C0 was 2.05 MPa. The interlayer splitting tensile strength of layered concrete at 20, 30, and 40 °C was 1.63 (T20), 1.18 (T30), and 0.83 MPa (T40), which was 21%, 44%, and 56% lower than bulk concrete (C0). In general, high temperature is harmful to interlayer splitting tensile strength of concrete.

#### 3.1.2. Relative Permeability Coefficient

Figure 8 shows the change of relative permeability coefficient of concrete at different temperatures. The relative permeability coefficients of layered concrete were larger than that of the bulk concrete. In addition, with the increase of temperature, the relative permeability coefficient of concrete shows an upward trend. Meanwhile, compared with C0, the relative permeability coefficients of T20, T30, and T40 are 1300, 1680, and 2320 times the bulk concrete, respectively. In general, the interlayer impermeability gradually deteriorated with the increase of temperature. The main reason for this phenomenon may be that the rise of temperature accelerates the evaporation of water in concrete. The lack of water reduces the hydration degree of cementitious materials and roughens the pore structure of lower concrete. The increase and interconnection of harmful pores eventually lead to the increase of the relative permeability coefficient [15].

#### 3.1.3. Resistance to Chloride Ion Penetration

The results of electric flux test are shown in Figure 9. The first rule obtained was that electric flux of layered concrete (T20, T30, and T40) was higher than that of bulk concrete (C0). Furthermore, the electric flux increased with the increase of temperature. For example, the electric flux of layered concrete at 20, 30, and 40 °C (T20, T30, and T40) was 10%, 26%, and 30% higher than that of the bulk concrete (C0), respectively. In general, the interlayer resistance to chloride ion penetration deteriorates with the increase of temperature.

### 3.2. Interlayer Bonding Quality Control Method

At the dam construction site, it is necessary to determine the decisive parameters affecting the interlayer bonding quality. Then establish the relationship between the parameters and the interlayer mechanical properties, impermeability, and chloride ion permeability. Thus, the judgment basis of interlayer bonding quality suitable for the construction site is obtained.

The closed-loop control system of dam interlayer bonding quality includes problem, perception, analysis, control, and objective (see Figure 10). In the process of dam pouring, the problem is that the bad environment leads to the deterioration of concrete interlayer performance. In order to solve this problem, it is necessary to determine the perception parameter and perception method. In this study, the equivalent age of the lower concrete was determined as the perception parameter. Finally, the interlayer bonding quality of the dam is made to meet the design requirements by controlling the equivalent age.

In order to effectively control the key parameters of the dam pouring quality, it is necessary to monitor the concrete temperature in real time. The main functions of the dam interlayer bonding quality control system include: (1) information monitoring; (2) when the key parameter exceeds the warning value, the system will automatically push the warning information to the construction personnel, supervisors and owners, and propose effective treatment measures—in addition, a real-time feedback mechanism needs to be established to close the early-warning information; and (3) statistical analysis of early-warning information.

#### 3.2.1. Data Collection

Data acquisition is the basis of the dam interlayer bonding quality control system. The data acquisition function can provide basic data guarantee for information query, information processing, and operation analysis in the system. Firstly, the collected original data are preliminarily screened to eliminate invalid data and duplicate data. In addition, duplicate data submitted due to hardware logic problems or acquisition program problems should be filtered, such as the same data submitted by the same machine at the same time point.

The data acquisition module is mainly based on the data of concrete initial temperature, pouring temperature, interval time, and water content in the intelligent construction information management platform of the dam project. In addition, in order to deal with the construction risks that may be faced during the concrete pouring process, it is also necessary to monitor meteorological information, such as ambient temperature, relative humidity, wind speed, and light intensity. Meteorological information is mainly monitored by weather stations (see Figure 11a). In the concrete vibrating process (see Figure 11b), it is necessary to monitor the paving thickness and vibration quality to prevent the phenomenon of leakage and under-vibration.

Data acquisition needs to comply with the following requirements: (1) data collection needs to be real-time, accurate, and comprehensive; (2) automatic data collection is independent and complete, and can be operated directly without other system modules; and (3) the automatic data collection system must back up the original data collected and ensure that data will not be lost when there is a problem with the network.

#### 3.2.2. Early-Warning Threshold

Freiesleben [33] proposed an equivalent age calculation method (Equation (5)) based on the Arrhenius Equation (4).
(4)$$K_{T}=A\cdot \exp\left(-\frac{E_{a}}{RT}\right)$$
where $K_{T}$ is the reaction rate constant; A is the frequency coefficient; $E_{a}$ is the apparent activation energy (kJ/mol), when the temperature is greater than or equal to 20 °C, $E_{a}=33.5\;\mathrm{kJ}/\mathrm{mol}$; R is the gas constant ($8.314\;J\cdot {\mathrm{mol}}^{-1}\cdot K^{-1}$); and T is the thermodynamic temperature (K, 273 °C).
(5)$$t_{e}=\int_{0}^{t}\exp\left[\frac{E_{a}}{R}\cdot \left(\frac{1}{T_{r}}-\frac{1}{T}\right)\right]$$
where $t_{e}$ is equivalent age; and $T_{r}$ is the absolute temperature at 20 °C (293 K). When calculating the equivalent age of the lower concrete, the upper limit of the integral is the interval between layers.

(1)The Relationship between Equivalent Age and Strength Coefficient

The strength coefficient of concrete can be calculated according to Equation (6). Since the bulk concrete has no interlayer interval, its equivalent age is 0 and the strength coefficient is 1. The relationship curve between the equivalent age of concrete and the strength coefficient is shown in Figure 12.
(6)$$F/F_{0}=A_{s}$$
where F is the interlayer splitting tensile strength; $F_{0}$ is the strength of bulk concrete; and $A_{s}$ is the strength coefficient [23].
(7)$$A_{s}=-0.04\cdot t_{e}+1\;\left(R^{2}=0.986\right)$$

The strength coefficient of low-heat cement concrete is linearly negatively correlated with the equivalent age, and the correlation coefficient is 0.986.

According to the relationship between the equivalent age and the strength coefficient, the strength coefficient can be controlled to 0.8, 0.7, and 0.6, respectively, and the corresponding equivalent ages are 4.7, 7.1 and 9.7 h. This indicates that if the interlayer splitting tensile strength is to be controlled within 80%, 70%, or 60% of the strength of bulk concrete, the equivalent age of the lower layer concrete should be controlled within 4.7, 7.1, or 9.7 h, respectively. If the equivalent age exceeds the corresponding control value, the interlayer bonding strength will not meet the control requirements.

(2)The Relationship between the Equivalent Age and the Relative Permeability Coefficient Ratio

The relative permeability coefficient ratio can be calculated according to Equation (8):(8)$$K_{r}/K_{r0}=K_{rr}$$
where $K_{r}$ is the relative permeability coefficient of layered concrete; $K_{r0}$ is the relative permeability coefficient of the bulk concrete; and $K_{rr}$ is the relative permeability coefficient ratio.

According to the relationship between the equivalent age and the relative permeability coefficient ratio (see Figure 13), the relative permeability coefficient ratio can be controlled to 500, 1000, and 1500, respectively, and the corresponding equivalent ages are 2.3, 5.4, and 8.6 h. This indicates that if the relative permeability coefficient between layers is controlled to be 500, 1000, or 1500 times of the relative permeability coefficient of the bulk concrete, the equivalent age of the lower layer concrete should be controlled within 2.3, 5.4, or 8.6 h, respectively.

(3)The Relationship between the Equivalent Age and the Electric Flux Ratio

There are two methods to test the chloride ion penetration resistance of concrete, which are the fast chloride ion migration coefficient method [34,35] and the electric flux method [36]. Electric flux is a parameter to evaluate the resistance of concrete to chloride ion penetration. Chloride ions will move to the positive electrode through the concrete specimen under the action of voltage. Therefore, the chloride penetration resistance of concrete can be indirectly reflected by the charge of concrete in a specific time. The more charge passing through concrete per unit time, the worse the chloride penetration resistance of concrete.

The electric flux ratio can be calculated according to Equation (9):(9)$$C/C_{0}=C_{r}$$
where C is electric flux of layered concrete; $C_{0}$ is electric flux of bulk concrete; and $C_{r}$ is electric flux ratio.

According to the relationship between equivalent age and electric flux ratio (see Figure 14), the electric flux ratio can be controlled to 1.1, 1.2, and 1.3, respectively, and the corresponding equivalent ages are 4.6, 9.6, and 13.7 h. This indicates that if electrical flux of layered concrete is controlled to be 1.1, 1.2, or 1.3 times of electric flux of bulk concrete, the equivalent age of lower layer concrete should be controlled within 4.6, 9.6, or 13.7 h, respectively.

#### 3.2.3. Early-Warning System

Interlayer bonding strength, water permeability resistance, and chloride ion permeability resistance of dam concrete can be controlled in accordance with the relationships between the equivalent age and the strength coefficient, the relative permeability coefficient ratio, and the electric flux ratio (see Table 3).

The early-warning thresholds were determined according to the early-warning model established in Figure 9, Figure 10 and Figure 11. Then, the warning information that meets the warning threshold was pushed to the corresponding responsible person according to its importance (see Table 3). For example: (1) push the yellow warning information to the construction personnel on site; (2) push the orange warning information to the construction personnel and the supervisor, respectively; and (3) push the red warning information to construction personnel, supervisors, and owners, respectively. When the relevant responsible personnel receive the warning information, the concrete will be treated according to the treatment plan corresponding to the warning level. After the completion, the supervisor will close the warning information.

The early-warning system realizes the transmission, analysis (data anomaly analysis and change trend analysis), statistics (final equivalent age statistics and early-warning information quantity statistics), query, and other functions of pouring quality monitoring data. Among them, the trend analysis is to calculate the equivalent age according to the measured temperature of concrete and then draw the corresponding curve according to the relationship between the equivalent age and time. The reasons for the decrease of concrete interlayer bond strength can be found out by data analysis based on the change curve of equivalent age and environmental information. The final equivalent age statistics function can realize the statistics of the final equivalent age of each layer of concrete. The warning information quantity statistics function can collect statistics on the amount of warning information pushed in any period of time.

## 4. Conclusions

Based on the test results of interlayer mechanical properties, impermeability, and chloride ion permeability of concrete at different temperatures, the following conclusions are obtained:(1)Increasing ambient temperature will reduce the mechanical property of concrete. On the contrary, the relative permeability and electric flux increase with the increase of temperature. It shows that the mechanical properties, impermeability, and anti-chloride ion permeability of concrete will deteriorate with the increase of temperature;(2)The equivalent age of lower concrete has a good linear correlation with strength coefficient, relative permeability coefficient ratio; and electric flux ratio. The equivalent age has a linear negative correlation with strength coefficient, and the correlation coefficient is 0.986. The equivalent age is linearly positively correlated with relative permeability coefficient ratio and the electric flux ratio, and correlation coefficients are 0.973 and 0.924, respectively; and(3)It is feasible to use the equivalent age to control the interlayer bonding quality of dam concrete. This method can effectively monitor the quality parameters of dam concrete. In addition, the dam interlayer bonding quality early-warning system can effectively deal with the possible construction risks and ensure the interlayer bonding quality.

The interlayer bonding quality control method has been applied in the dam construction process. This method is effective for the control of interlayer bonding quality. In the future, it is planned to introduce the influence factors of humidity and wind speed into the calculation equation of equivalent age to deal with the interlayer bonding quality control of concrete under complex environmental conditions.

## Figures and Tables

**Figure 1 materials-14-05192-f001:**
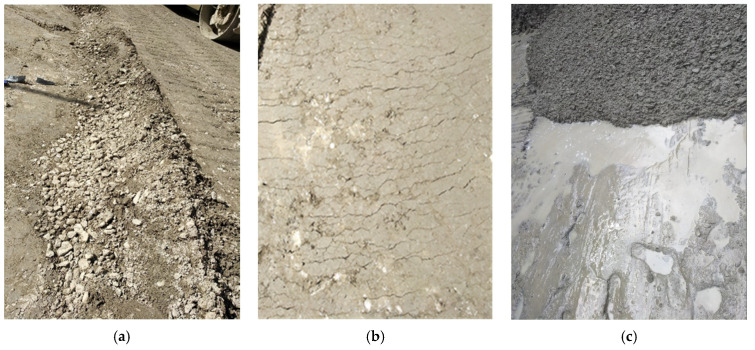
Deterioration of concrete layer performance. (**a**) Aggregate whitening, (**b**) dry shrinkage and cracking, and (**c**) bleeding (normal concrete).

**Figure 2 materials-14-05192-f002:**
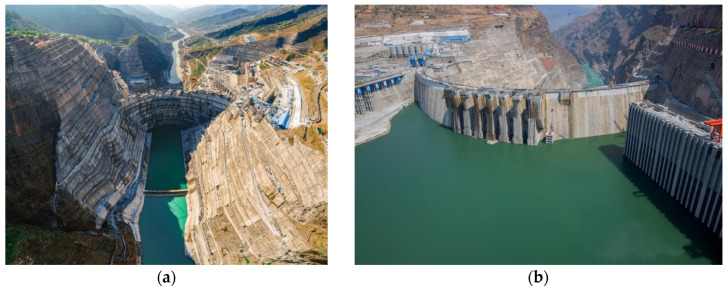
(**a**) Wudongde dam, (**b**) Baihetan dam.

**Figure 3 materials-14-05192-f003:**
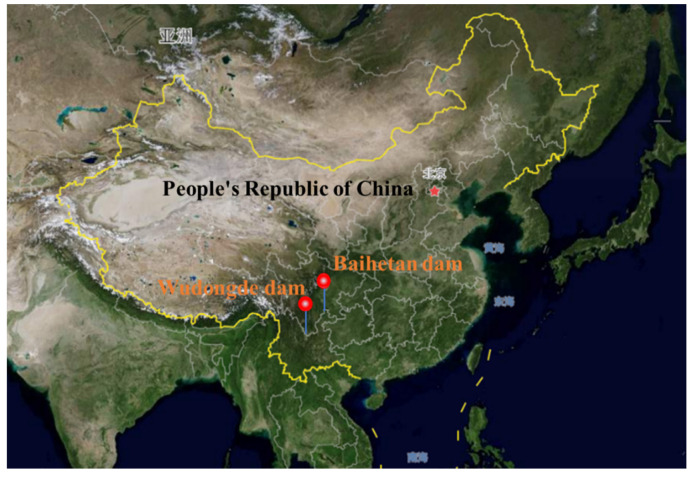
The location of the dams.

**Figure 4 materials-14-05192-f004:**
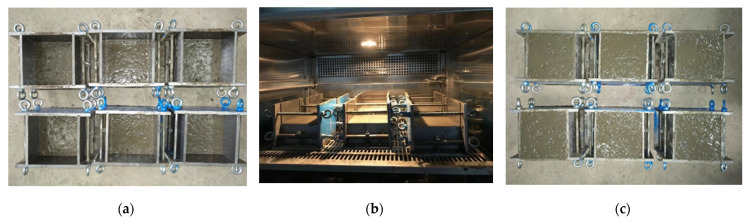
(**a**) Pouring lower layer concrete; (**b**) putting in the environmental chamber, and (**c**) pouring upper layer concrete.

**Figure 5 materials-14-05192-f005:**
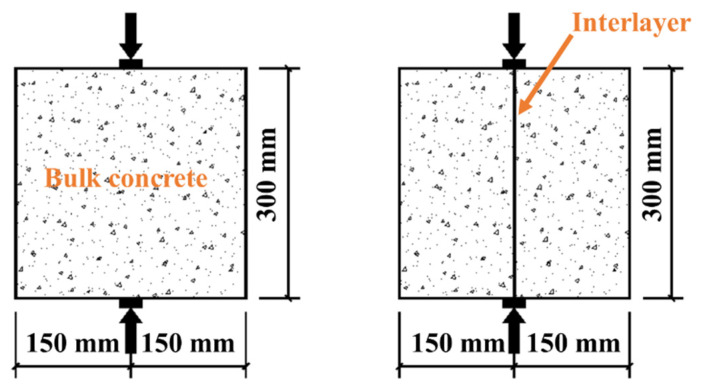
Splitting tensile strength test of concrete.

**Figure 6 materials-14-05192-f006:**
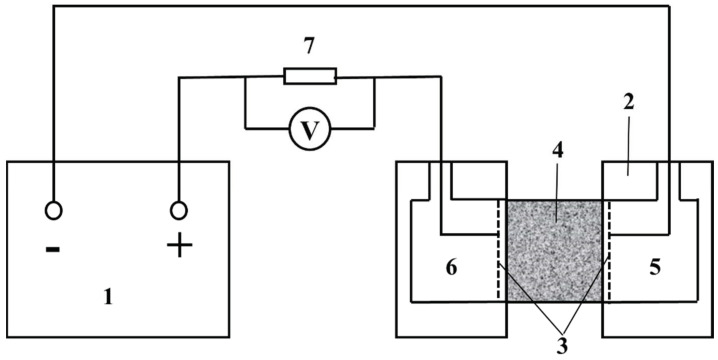
Schematic diagram of electric flux test device. 1. DC power supply. 2. Test tank. 3. Rubber mat. 4. Concrete specimen. 5. Sodium chloride solution (concentration 3%). 6. Sodium hydroxide solution (0.3 mol/L). 7. Electric resistance.

**Figure 7 materials-14-05192-f007:**
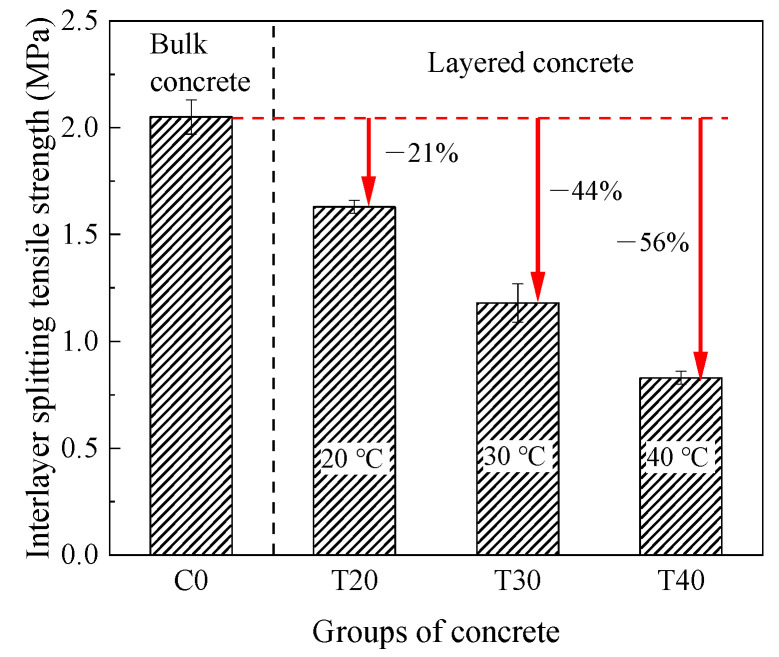
Splitting tensile strength test results.

**Figure 8 materials-14-05192-f008:**
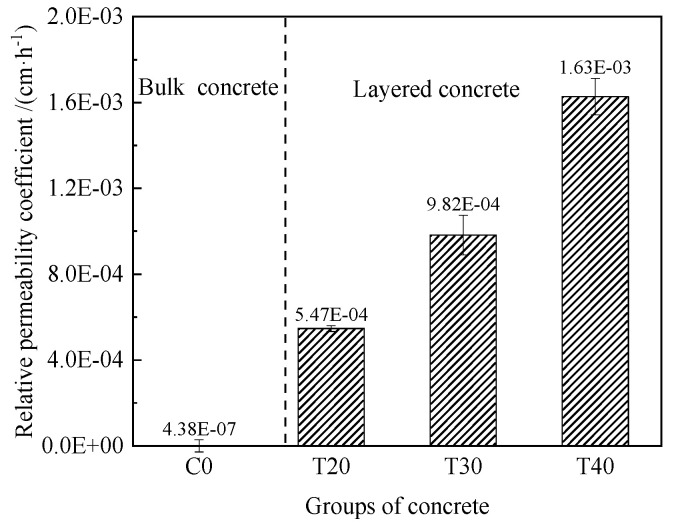
Relative permeability coefficient test results.

**Figure 9 materials-14-05192-f009:**
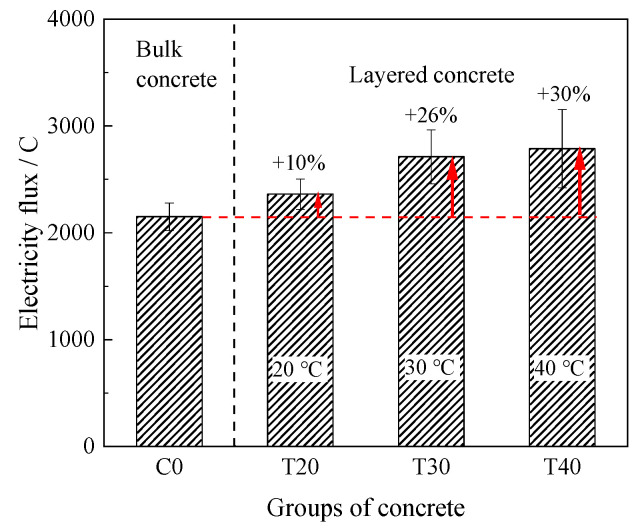
Electric flux test results.

**Figure 10 materials-14-05192-f010:**
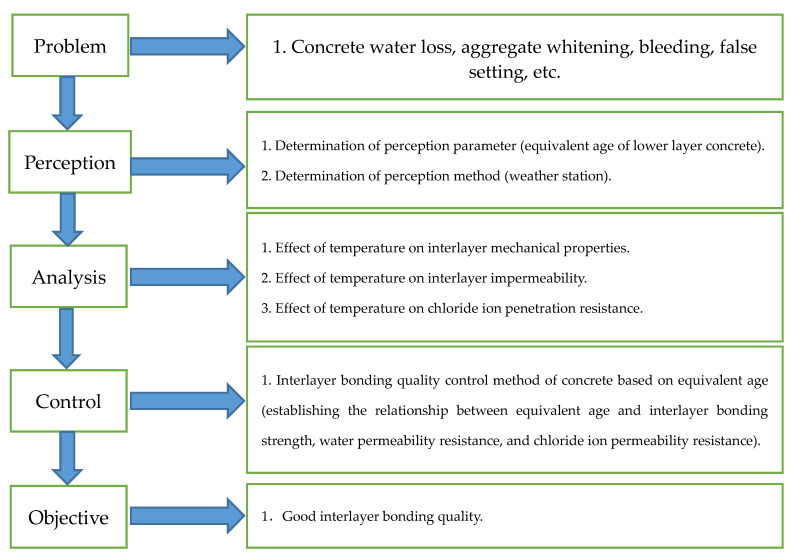
Interlayer bonding quality closed-loop control system.

**Figure 11 materials-14-05192-f011:**
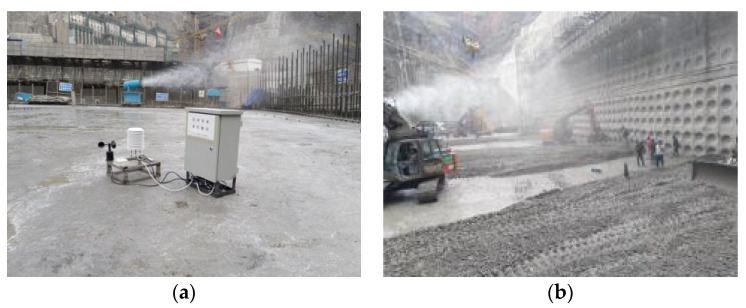
(**a**) Small weather stations, and (**b**) concrete paving and vibration.

**Figure 12 materials-14-05192-f012:**
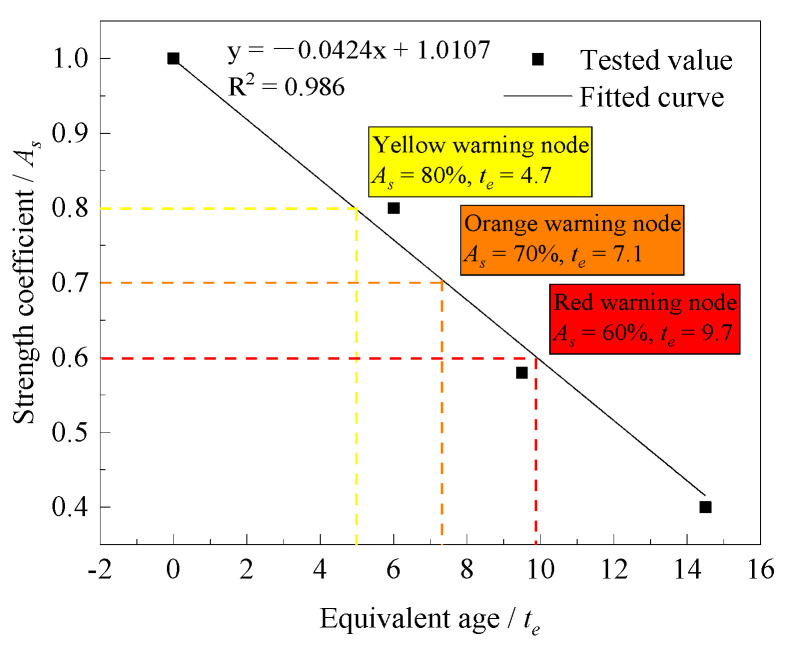
Relationship between the equivalent age and the strength coefficient.

**Figure 13 materials-14-05192-f013:**
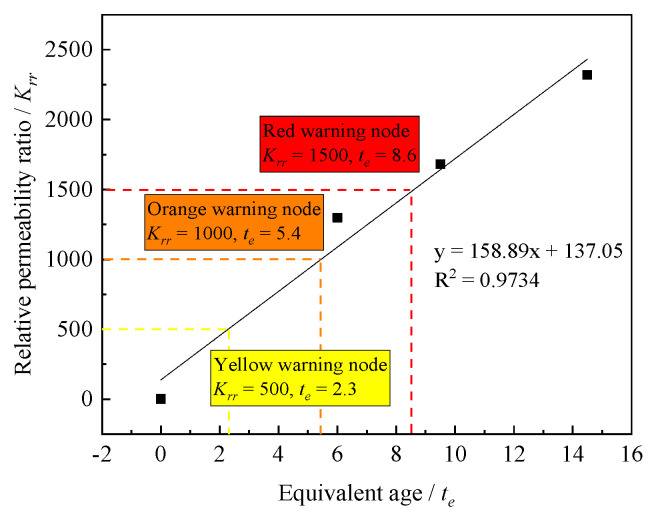
Relationship between the equivalent age and the relative permeability coefficient ratio.

**Figure 14 materials-14-05192-f014:**
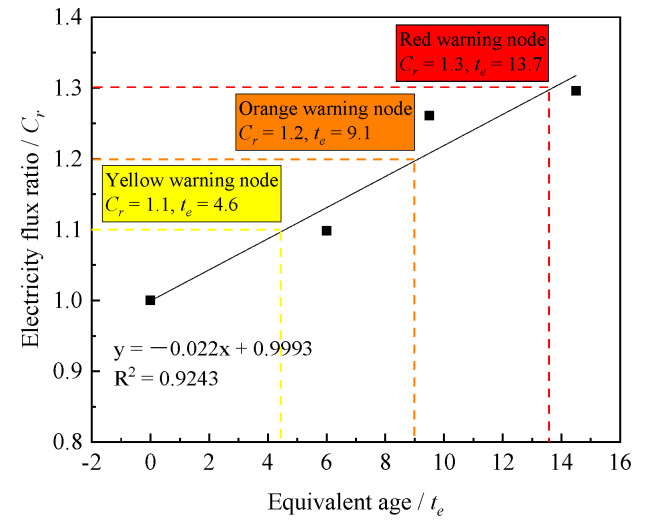
Relationship between the equivalent age and the electric flux ratio.

**Table 1 materials-14-05192-t001:** Properties of cement and fly ash.

Properties		Cement	Fly Ash
Chemical composition (% by mass)	SiO_2_	23.24	50.6
	Al_2_O_3_	4.07	27.1
	Fe_2_O_3_	3.02	7.1
	CaO	61.64	4.5
	MgO	4.74	1.2
	SO_3_	2.05	0.3
	Alkali content	0.45	1.3
	Loss on ignition	1.08	4.4
Mineral composition (% by mass)	C_3_S	32.4	-
	C_2_S	47.5	-
	C_3_A	0.2	-
	C_4_AF	14.6	-
	Gypsum	3.5	-
Physical properties	Specific gravity (g/cm^3^)	3.20	1.8
	Specific surface (m^2^/kg)	319	-
Compressive strength (MPa)	7 days	21.3	-
	28 days	48.4	-

**Table 2 materials-14-05192-t002:** Mix proportion of concrete (kg/m^3^).

W/B	Cement	Fly Ash	Artificial Sand	Coarse Aggregate	Super-Plasticizer	Air-Entraining Agent
0.48	100	123	777	1323	1.596	0.0798

**Table 3 materials-14-05192-t003:** Early-warning threshold of interlayer bonding quality.

Types of Early Warning	Control Parameters	Control Index	Corresponding Equivalent Age	Early-Warning Level	Treatment Measures
Early warning of interlayer bonding strength	Strength coefficient	0.8	4.7 (Equivalent age > 4.7 h, push early-warning information to construction personnel)	Yellow	Increase spray volume
		0.7	7.1 (Equivalent age > 7.1 h, push early-warning information to construction personnel and supervisors)	Orange	Cover heat insulating material
		0.6	9.7 (Equivalent age > 9.7 h, push early-warning information to construction personnel, supervisors, and owners)	Red	Surface remodeling or treated as cold joint
Early warning of interlayer impermeability	Relative permeability ratio	500	2.3 (Equivalent age > 2.3 h, push early-warning information to construction personnel)	Yellow	Increase spray volume
		1000	5.4 (Equivalent age > 5.4 h, push early-warning information to construction personnel and supervisors)	Orange	Cover heat insulating material
		1500	8.6 (Equivalent age > 8.6 h, push early-warning information to construction personnel, supervisors, and owners)	Red	Surface remodeling or treated as cold joint
Early warning of interlayer resistance to chloride ion penetration	Electric flux ratio	1.1	4.6 (Equivalent age > 4.6 h, push early-warning information to construction personnel)	Yellow	Increase spray volume
		1.2	9.6 (Equivalent age > 9.6 h, push early-warning information to construction personnel and supervisors)	Orange	Cover heat insulating material
		1.3	13.7 (Equivalent age > 13.7 h, push early-warning information to construction personnel, supervisors, and owners)	Red	Surface remodeling or treated as cold joint

## Data Availability

The data presented in this study are available on request from the corresponding author.

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
