# Peer review of "Research on Interlayer Bonding Quality Control Method of Dam Concrete Based on Equivalent Age"

_materials, 2021, doi:10.3390/ma14185192_

Round 1

Reviewer 1 Report

In my opinion, the manuscript is well written, the topic is interesting, but I found several typos and errors that should be corrected to improve the overall quality of the work. From my analysis I found several specific issues that I’d like to address:

  • In the abstract and the whole manuscript, it is not clearly stated what it is meant by “increased temperature”. If it is curing temperature during hardening or if it is some sort of external parameter that affects the hardened concrete. This specific parameter should be well defined within the whole text (the increase of curing temperature?). It is not clear.
  • Introduction: In the first paragraph, there is an overuse of the word “intelligent”, which makes it more challenging to read. I would propose to rewrite this part to improve the flow of the text.
  • Ustabas[29] – There should be a space between the name and reference. 2780 kg/m3 – There are several similar typos within the text. Please, check the text carefully.
  • Table 1: It should be provided in the text how the presented properties were determined. The unit of the specific gravity is missing in the table.
  • The compressive strength after 28 days seems a bit strange. According to most of the standards, there are three main types of cement as 32.5; 42.5, and 52.5, and their respective compressive strength should be at least the same as the type or usually higher. Here, 31.7 seems to be too low. The type of selected cement should be described more in detail.
  • The Chemical and mineral composition of fly ash are missing.
  • It is not clear what is artificial sand and coarse aggregate. What kinds were used?
  • How many samples were prepared for each analysis?
  • Figure 3: There is a typo in “bulk” Additionally, according to my experience, the term “ordinary” is more often used to describe plain concrete without any reinforcement.
  • Figure 7: There is a typo in the world Perception (left side, under Problem).
  • 2. Interlayer bonding quality control method: In this full chapter, there is a lack of references.
  • In my opinion, Table 3 is too generic.
  • The conclusions should be improved. It is provided as a list of the results without a summary of the proposed study or the aims of the study. What is the practical impact of the results? Future plans?
  • 9/34 self-citations seem high and should be fixed.

Reviewer 2 Report

In the introduction part you could add a paragraph regarding the influence of the different raw materials used as aggregates in construction applications such as concrete and how these play severe role on the final mechanical behaviour of these products and on interlayer bonding quality (i.e Petrounias et al. 2018, i.e. Jaskula et al. 2017).

Furthermore, you could also refere on how  the mechanical behaviour of concrete may be affected by the microstructure of the mineral raw materials used as concrete aggregates. The manuscript it was very intresting but you can change some points:

figures 2: i need one basic map..or something like a map or mapping with pictures a,b.

2.1 coarse aggregates? what kind of rock?sedimentary? any picture for aggregates?   water /cement ratio?

2.2 speciment preparetion. any picture for preparation?

2.3.3 electric flux methods needs more details. 

3.2 any more explain text in the beginning.  in my opinion i need more explain texts in the discussion parts such as 3.2.1 and 3.2.2. for data collection do you have any protocol?

figure 11 it is not correct because it is very poor the statistical data.it is only one basic relationship. conclusions are very specific and very good.          

Reviewer 3 Report

In this paper it is studied the interlayer splitting tensile strength, relative permeability coefficient, electric flux for concrete dams. For this, a comprehensive early warning and control system of dam interlayer bonding quality has been proposed. In this paper, there are some results very interesting as Fig. 6 and Fig. 11.

The paper is very interesting and current. The methodology, organization and academic level are good. Therefore, only some aspects should be improved:

1) Figs. 1-2: Where did you get the photos? Please, indicate the source.

2) Eq. (3). It is not clear how you correlate the electrical current with the choride diffusion. Please, explain it in the text. Reference a) maybe can help you. 

3) Fig. 7: please adjust it.

4) Fig. 11: It is very interesting. Is it a your correlation? Please, explain more details. To study the chloride diffusion by considering an "electrical filed" you can use the Nerst-Plank equation (see reference b)).

5) Some sections are very small (e.g., 2.1, 3.1.1, etc.). If possible, I suggest writing more. For nomenclature and models of the dams please see reference c).

Suggested references:

a) https://doi.org/10.1016/j.conbuildmat.2015.07.190

b) https://doi.org/10.1016/j.jobe.2020.101296

c) https://doi.org/10.1016/j.ijdrr.2021.102311

Round 2

Reviewer 2 Report

 Accept in present form